# Switching Latent Bandits

## Abstract

We consider a Latent Bandit problem where the latent state keeps changing in time according to an underlying Markov Chain and every state is represented by a specific Bandit instance. At each step, the agent chooses an arm and observes a random reward but is unaware of which MAB he is currently pulling. As typical in Latent Bandits, we assume to know the reward distribution of the arms of all the Bandit instances. Within this setting, our goal is to learn the transition matrix determined by the Markov process, so as to minimize the cumulative regret. We propose a technique to solve this estimation problem that exploits the properties of Markov Chains and results in solving a system of linear equations. We present an offline method that chooses the best subset of possible arms that can be used for matrix estimation, and we ultimately introduce the SL-EC learning algorithm based on an Explore Then Commit strategy that builds a belief representation of the current state and optimizes the instantaneous reward at each step. This algorithm achieves a regret of the order $\widetilde{\mathcal{O}}(T^{2/3})$ with $T$ being the considered horizon. We make a theoretical comparison of our approach with Spectral decomposition techniques. Finally, we illustrate the effectiveness of the approach and compare it with state-of-the-art algorithms for non-stationary bandits and with a modified technique based on spectral decomposition.

## 1 Introduction

The Multi-Armed Bandit (MAB) framework is a well-known model used for sequential decision-making with little or no information. This framework has been successfully applied in a large number of fields, such as recommender systems, advertising, and networking. In the general MAB formulation, a learner sequentially selects an action among a finite set of different ones. The choice over the arm to select is made by properly balancing the exploration-exploitation trade-off with the goal of maximizing the expected total reward over a horizon $T$ and guaranteeing the *no-regret* property, thus meaning that the loss incurred by not knowing the best arm is increasing sublinearly over time. Standard MAB literature requires the payoff of the available actions to be stationary (i.e., rewards come from a fixed distribution) in order to design efficient no-regret algorithms.

However, in many real-life applications, the stationarity assumption may not necessarily hold as data may be subjected to changes over time. In some applications, it is also possible to identify different data distributions each one corresponding to a specific working regime. In cases of large availability of historical data appearing in the form of past user interactions, it is possible to learn *offline* the observation models associated with the different arms for each working regime. Exploiting the knowledge on observation models leads to many advantages over the *fully online exploration* setting where no prior information is available at the beginning and a massive number of interactions is required to learn the observation models associated with each working regime. Even if the latent regime is not directly observable, by knowing the observation distributions, it is possible to learn the dynamics of the regimes from the interaction process and thus infer the current state at each step. Identifying the latent state accelerates the adaptation of the agent to the environment leading to improved performances over time.

Learning observation models independently and before transition models may be a possible choice when there is little availability of computational resources. Indeed, we will show in the following that spectral decomposition techniques, which are used to jointly learn the observation and the transition model, typically require a large number of samples and involve computationally intensive operations. Other scenarios where

we can assume that the observation models are already known are those where the models are learned offline from samples generated by simulators. Once these models are deployed in a non-stationary environment, the dynamics can be learned by interacting with the environment. We can consider for example the problem of resource allocation such as the electricity allocation in a specific residential area. Each arm represents a specific allocation configuration, while the rewards represent the extent to which the allocation has been optimal. Obviously, this optimality of the allocation depends on the state of the system, which may be conditioned by several factors such as environmental conditions, community trends, seasonality.

Another possible scenario that suits our setting is the one of *Transfer Learning*, where partial knowledge of the system (in our case the observation model) can be used in a different context where dynamics are different (and new transition models need to be learned). In the scenario previously considered, we can consider using the same observation models in a new residential area, with a structure analog to the first one (thus justifying the use of the same observation model) but located in a different place, with potentially different weather conditions and inhabitants having different behaviors.

Assuming the existence of a finite set of discrete latent states is a relevant choice when approaching the modeling of complex real-life problems characterized by different and recurrent working regimes. These regimes can be typically observed in domains such as the financial market and online advertising, typically marked by high volatility and specific seasonality patterns (M. et al., 2022; Heston & Sadka, 2008; Guo et al., 2021). Introducing a more practical example, in the stock exchange market where different models are available, typically one for each regime, it is relevant to choose the best stock to exchange based on the unknown market condition. The different regimes may be identified through the availability of past data by either considering some seasonality patterns or specific indicators of the market conditions using the domain knowledge of experts. In this case, inferring the current state of the market, associating a duration with it, and predicting future transitions allows taking fairer decisions and bringing higher outcomes.

Past works focused on this state identification problem under the assumption of knowing the conditional observation models (Maillard & Mannor, 2014; Zhou & Brunskill, 2016) and defined theoretically optimal UCB algorithms. Follow-up work by Hong et al. (2020a) provided more practical Thompson Sampling algorithms also considering the problem of model misspecification and came up with an analysis on the Bayes regret.

The works cited above assume that the latent state does not change during the interaction process: once the real state is identified, the agent can act optimally. Differently, in this work, we embrace a more realistic scenario and assume that the latent state can change through time. In accordance with the latent bandits setting, we assume that the learning agent is aware of the observation models of the arms conditioned on each latent state. A setting similar to ours has been considered also in Hong et al. (2020b), the key difference is that they assume to have either full or partial knowledge of both the observation model and the transition model. We instead focus on the more challenging problem of learning the transition model given the knowledge of the observation models and maximizing the cumulative reward over $T$ interaction steps. More specifically, our problem is modeled by assuming the existence of a finite set $\mathbb{S}$ of different MABs all sharing the same set of finite arms $\mathbb{I}$, each generating rewards (observations) in a finite set $\mathbb{V}$. Each state $s \in \mathbb{S} = \{s_1, \ldots, s_S\}$ represents a different instance of a MAB. At each time step $t$, there is a transition from latent state $s_{t-1}$ to the new latent state $s_t$ according to the transition matrix governing the process. The action $a_t$ selected in $t$ will thus generate a reward conditioned on the latent state $s_t$. Assuming the transition dynamics to be described using a Markov Chain can be advantageous for modeling durations of states that can be represented with geometric distributions.

**Our Contribution**   We summarize here the main aspects and contributions related to this work:

- we design a procedure for the estimation of the transition matrix that converges to the true value under some mild assumptions. In order to obtain this result, we exploit the information derived from the conditional reward models, and we use some properties of Markov Chains;

- we provide high-probability confidence bounds for the proposed procedure using known results from statistical theory and novel estimation bounds of samples coming from Markov Chains;

- we propose the *Switching Latent Explore then Commit* (SL-EC) algorithm that uses the presented estimation method and then exploits the learned information achieving a $\widetilde{\mathcal{O}}(T^{2/3})$ regret bound on a finite horizon $T$;

- we illustrate the effectiveness of the approach and compare it with state-of-the-art algorithms for the non-stationary bandits setting.

## 2 Related Works

**Non-stationary Bandits**   Non-stationary behaviors are closer to real-world scenarios, and this has induced a vast interest in the scientific community leading to the formulation of different methods that consider either abruptly changing environments (Garivier & Moulines, 2011), smoothly changing environments (Trovò et al., 2020), or settings with a bounded variation of the rewards (Besbes et al., 2014). It is known that when rewards may arbitrarily change over time, the problem of Non-Stationary Bandits is intractable, meaning that only trivial bounds can be derived on the dynamic pseudo-regret. That is the main reason why in the literature there is a large focus on non-stationary settings enjoying some specific structure in order to design algorithms with better guarantees. Non-stationary MAB approaches typically include both passive methods in which arm selection is mainly driven by the most recent feedback (Auer et al., 2019; Besbes et al., 2014; Trovò et al., 2020) and active methods where a change detection layer is used to actively perceive a drift in the rewards and to discard old information (Liu et al., 2017; Cao et al., 2018). Works such as Garivier & Moulines (2011) provide a $\mathcal{O}(\sqrt{T})$ regret guarantee under the assumption of knowing the number of abrupt changes. Other works, such as Besbes et al. (2014), employ a fixed budget to bound the total variation of expected rewards over the time horizon. They are able to provide a near-optimal frequentist algorithm with pseudo-regret $\mathcal{O}(T^{2/3})$ and a distribution-independent lower bound. All the above methods are not suited for environments that switch between different regimes as they do not keep in memory past interactions but rather tend to forget or discard the past.

A particular type of non-stationary Bandit problem related to our work includes the *restless Markov* setting (Ortner et al., 2014; Slivkins & Upfal, 2008) where each arm is associated with a different Markov process and the state of each arm evolves independently of the learner's actions. Differently, Fiez et al. (2018) investigate MAB problems with rewards determined by an unobserved Markov Chain where the transition to the next state depends on the action selected at each time step, while Zhou et al. (2021) focus on MAB problems where the state transition dynamics evolves independently of the chosen action. This last work has many similarities with our setting. The main difference lies in the fact that they do not assume to know the conditional reward models and learn them jointly with the transition matrix. They make use of spectral decomposition techniques (Anandkumar et al., 2014) and use this tool in a regret minimization algorithm achieving a $\mathcal{O}(T^{2/3})$ regret bound. Their setting is more complex than ours but involves further assumptions, like the invertibility of the transition matrix that defines the Chain. Furthermore, spectral methods need a vast amount of samples in order to provide reasonable estimation errors and can hardly be used in large problems. A detailed discussion on the differences between the work of Zhou et al. (2021) and ours will be presented in Section 5.2.

**Latent Bandits**   More similar lines of work are related to bandit studies where latent variables determine the distribution of rewards (Maillard & Mannor, 2014; Zhou & Brunskill, 2016). In these works, the unobserved state is fixed across different rounds and the conditional rewards depend on the latent state. Maillard & Mannor (2014) developed UCB algorithms without context considering the two different cases in which the conditional rewards are either known or need to be estimated. This line of work has been extended to the contextual bandit case in Zhou & Brunskill (2016) where there is an offline procedure to learn the policies and a selection strategy to use them online. Hong et al. (2020a) proposed a TS procedure in the contextual case that updates a prior probability over the set of states in order to give a higher probability to the real latent state. A non-stationary variant of this setting is proposed in Hong et al. (2020b) where the latent states are assumed to change according to a Markov Chain. They develop TS algorithms under different cases when both the reward and transition models are completely known and when partial information about them is available. For the partial information case, they provide an algorithm based on particle filter which will be used for comparison in the experimental section. Differently from Hong et al. (2020b), we do not

assume any prior information about the transition matrix and we learn it through interactions with the environment using the information about the reward models.

Another interesting work associated with latent bandits is the one from Kwon et al. (2022) where, differently from previously cited works, they assume an episodic setting with a fixed horizon $H$. At the beginning of each episode, a specific MAB instance is sampled from a fixed mixing distribution and the agent interacts with the sample MAB until the end of the episode, without being aware of the MAB she is interacting with. The goal is to learn both the mixture weights and the reward distributions associated with each MAB. The relevant differences with our work rely on the episodic setting, while we assume a non-episodic one, and on the fact that in Kwon et al. (2022), MABs are sampled independently at the beginning of each episode, while in our case there is a dependence between MABs that switch potentially at every time step based on the underlying Markov process. Another main difference with the previously considered works is that they provide results in terms of sample complexity needed in order to learn a near-optimal policy, not taking into account the suffered regret.

## 3 Switching Latent Bandits

### 3.1 Preliminaries

**Markov Chains** A Markov Chain (or Markov Process) (Feller, 1968) over the state space $\mathbb{S}$ is a stochastic process $(S_t)_{t=1}^{\infty}$ satisfying the Markov property, meaning that for all $s_i, s_j \in \mathbb{S}$ and $t > 0$:

$$P(S_{t+1} = s_j | S_t = s_i, \ldots, S_0 = s_0) = P(S_{t+1} = s_j | S_t = s_i).$$

More formally, a Markov chain is identified by a tuple $\langle \mathbb{S}, \boldsymbol{P}, \boldsymbol{\nu} \rangle$ with $\mathbb{S} = \{s_1, \ldots, s_S\}$ being a (finite) set of states, $\boldsymbol{P}$ is a state transition probability matrix with element $P_{ss'} = P(S_{t+1} = s' | S_t = s)$ and $\boldsymbol{\nu} \in \Delta^{|\mathbb{S}|-1}$ is the initial state distribution with $\nu_s = P(S_0 = s)$. Given the starting distribution $\boldsymbol{\nu}$ and the transition matrix $\boldsymbol{P}$, we can define the probability distribution over the state space after $n$ steps as:

$$\boldsymbol{\nu}^{(n)} = \boldsymbol{\nu} \boldsymbol{P}^n.$$

We can classify Markov Chains according to the different properties they satisfy. In particular, a Markov Chain is *Regular* if some power $n$ of the transition matrix $\boldsymbol{P}^n$ has only positive elements (Puterman, 1994). If a Markov Chain is Regular, it admits a unique stationary distribution, as can be seen in the following:

**Proposition 3.1.** *Let $\boldsymbol{P}$ be the transition matrix of a Regular Markov Chain and $\boldsymbol{v}$ an arbitrary probability vector. Then:*

$$\lim_{n \to \infty} \boldsymbol{v} \boldsymbol{P}^n = \boldsymbol{\pi},$$

*where $\boldsymbol{\pi}$ is the unique stationary distribution of the chain, and the components of the vector $\boldsymbol{\pi}$ are all strictly positive.*

Having established the concept of stationary distribution, we give now another core definition, the one of *spectral gap*, that will be useful for what will follow. Before that, we define the set $(\lambda_i)_{i \in [S]}$ of ordered eigenvalues of $\boldsymbol{P}$, with $1 \geq |\lambda_1| \geq |\lambda_2| \geq \cdots \geq |\lambda_S|$. Assuming to consider a Regular Markov Chain, the system has a unique stationary distribution, and an eigenvalue $\lambda_1 = 1$.

**Definition 3.1.** *The spectral gap $\beta$ of a Markov Process defined by transition matrix $\boldsymbol{P}$ is $1 - |\lambda_2|$.*

The spectral gap provides valuable information about the process. For Regular Markov Chains, the spectral gap controls the rate of exponential decay to the stationary distribution (Saloff-Coste, 1997).

### 3.2 Problem Formulation

Consider a finite set $\mathbb{S}$ of $S = |\mathbb{S}|$ different MAB problems. Each MAB has a finite set of discrete arms $\mathbb{I} := \{a_1, \ldots, a_I\}$ with cardinality $I = |\mathbb{I}|$ and, by pulling an arm $a$, it is possible to get a reward $r$ taken from the set $\mathbb{V} = \{r_1, \ldots, r_V\}$ of possible rewards. In our setting, we assume to have a finite set of rewards $V = |\mathbb{V}|$ with each reward $r \in \mathbb{V}$ bounded for simplicity in the range $[0, 1]$. All the considered MABs share

the same sets of arms $\mathbb{I}$ and rewards $\mathbb{V}$. At each step, the MABs alternate according to an underlying Markov Chain having transition probability $\boldsymbol{P}$ with size $S \times S$.

The interaction process is as follows: at each time instant $t$, the agent chooses an arm $I_t = a$ and observes a reward $R_t = r$ that is determined by the underlying state $S_t = s$ of the process. More formally, the distribution associated with the revealed reward is

$$Q(r|s,a) := P(R_t = r|S_t = s, I_t = a). \tag{1}$$

For the moment, we will stick with the assumption that the distribution $Q(\cdot|s,i)$ over possible rewards is categorical. In Section 5.1, we will see how continuous distributions can also be handled in this setting. Given all the MABs, the actions and possible observations, we can define the three-dimensional observation tensor $\mathbf{O}$ with size $S \times I \times V$ where the element $O_{s,a,r}$ represents the probability of observing the reward $r$ being in state $s$ and pulling arm $a$.

In particular, by fixing a state $s$ and an action $a$, the vector $\mathbf{O}_{s,a,:}$ contains the parameters of the categorical distribution associated with state $s$ and action $a$. Motivated by the realistic scenario of massive availability of past interaction data in domains such as recommender systems that allow learning the reward models during an offline phase, we make the assumption of knowing the observation tensor $\mathbf{O}$ while our objective is to learn the transition matrix $\boldsymbol{P}$ that governs the Chain.

For what follows, it will be useful to represent the information contained in the observation tensor $\mathbf{O}$ by using a new matrix $\boldsymbol{O}$ with size $IV \times S$, that we call action observation matrix. Precisely, we encode the pair $(a,r)$ of action and reward into a variable $d \in \{1, 2, \ldots, IV\}$. Then, for any pair $(a,r)$ and its corresponding mapping $d$, and any state $s$, we have:

$$\boldsymbol{O}(d,s) = Q(r|s,a) = \mathbf{O}_{s,a,r}, \tag{2}$$

where variable $d$ represents the $d$-th row of the action observation matrix $\boldsymbol{O}$.

### 3.3 Reference Matrix Definition

We will introduce here some elements whose utility will be clarified in Section 4.

Let's consider the set $\mathbb{C}_{\mathbb{S}} := \{(s_i, s_j)|s_i, s_j \in \mathbb{S}\}$ with $|\mathbb{C}_{\mathbb{S}}| = S^2$ of all the ordered combinations of pairs of states. These combinations identify all the possible state transitions that can be seen from a generic time step $t$ to the successive one $t+1$. Analogously, we can define the sets $\mathbb{C}_{\mathbb{I}} := \{(a_i, a_j)|a_i, a_j \in \mathbb{I}\}$ with $|\mathbb{C}_{\mathbb{I}}| = I^2$ and $\mathbb{C}_{\mathbb{V}} := \{(r_i, r_j)|r_i, r_j \in \mathbb{V}\}$ with $|\mathbb{C}_{\mathbb{V}}| = V^2$ which are respectively the ordered combinations of pairs of all consecutive arms and of consecutive rewards that can be seen in two contiguous time intervals. From the knowledge of the observation tensor $\mathbf{O}$ and for each $(s_i, s_j) \in \mathbb{C}_{\mathbb{S}}, (a_i, a_j) \in \mathbb{C}_{\mathbb{I}}, (r_i, r_j) \in \mathbb{C}_{\mathbb{V}}$, we are able to compute the following probabilities:

$$P(R_t = r_i, R_{t+1} = r_j|S_t = s_i, S_{t+1} = s_j, I_t = a_i, I_{t+1} = a_j) = O_{s_i,a_i,r_i} O_{s_j,a_j,r_j}. \tag{3}$$

Equation 3 basically allows us to define the probability associated with each possible couple of rewards, given each couple of actions and each couple of states that can occur in consecutive time steps. Hence, by fixing a specific combination of arms $(a_h, a_k)$ from $\mathbb{C}_{\mathbb{I}}$ and by leveraging Equation 3, we can build matrix $\boldsymbol{H}^{a_h,a_k} \in \mathbb{R}^{V^2 \times S^2}$ where the elements along the rows are associated to combinations in $\mathbb{C}_{\mathbb{V}}$ and the elements along the columns are associated to combinations in $\mathbb{C}_{\mathbb{S}}$. The generic element $H^{a_h,a_k}_{d,e}$ contains the value computed in Equation 3 associated to the d-th combination of rewards in $\mathbb{C}_{\mathbb{V}}$ and the e-th combination of states in $\mathbb{C}_{\mathbb{S}}$ assuming to have pulled actions $(a_h, a_k)$. Having established this procedure to build matrix $\boldsymbol{H}^{a_h,a_k}$ for a couple of actions $(a_h, a_k)$, we can now build similar matrices associated with each of the other combinations of arms. By stacking all these matrices together, we get the matrix $\boldsymbol{A} \in \mathbb{R}^{I^2 V^2 \times S^2}$.

This matrix is a reformulation of the observation tensor $\mathbf{O}$ that expresses the relation between pairs of different elements. The procedure just described for the construction of matrix $\boldsymbol{A}$ can also be expressed in terms of the action observation matrix $\boldsymbol{O}$. Specifically, we have that:

$$\boldsymbol{A} = \boldsymbol{O} \otimes \boldsymbol{O},$$

where symbol $\otimes$ refers to the Kronecker product (Loan, 2000). We provide a simple table representation of the reference matrix in Appendix D. We report here a property of the Kronecker product that will be useful in the following. We have that:

$$\sigma_{\min}(\boldsymbol{A}) = \sigma_{\min}^2(\boldsymbol{O}). \tag{4}$$

The definition of the matrix $\boldsymbol{A}$ will be relevant to the proposed estimation method. In the following, we will refer to matrix $\boldsymbol{A}$ also with the name reference matrix.

### 3.4 Belief Update

As previously said, at each time step $t$, we only observe the reward realization, but we are unaware of the Bandit instance from which the arm has been pulled. However, it is possible to define a belief representation over the current state by using the information derived from the observation tensor $\mathbf{O}$ and the transition matrix $\boldsymbol{P}$ defining the Chain.

We need to introduce a belief vector $\mathbf{b}_t \in \Delta^{S-1}$ representing the probability distribution over the current state at time $t$. The belief update formulation will follow the typical correction and update step, where the correction step adjusts the current belief $\mathbf{b}_t$ using the reward $r_t$ obtained by pulling arm $a_t$ and the prediction step computes the new belief $\mathbf{b}_{t+1}$ simulating a transition step. The overall update is as follows:

$$\mathrm{b}_{s,t+1} = \frac{\sum_{s'} \mathrm{b}_{s',t} Q(R_t = r_t | S_t = s', I_t = a_t) \boldsymbol{P}(s|s')}{\sum_{s''} Q(R_t = r_t | S_t = s'', I_t = a_t) \mathrm{b}_{s'',t}}. \tag{5}$$

The choice of the arm to pull is driven, at each step $t$, by

$$I_t = \arg\max_{a \in \mathbb{I}} \sum_{s \in \mathbb{S}} \sum_{r \in \mathbb{V}} r Q(r|s,a) \mathrm{b}_{s,t}. \tag{6}$$

In this case, the goal is to pull the arm that provides the highest instantaneous expected reward, given the belief representation $\mathbf{b}_t$ of the states.

### 3.5 Assumptions

We need now to introduce some assumptions that should hold in our setting:

**Assumption 3.1.** *The smallest element of the transition matrix $\epsilon := \min_{i,j \in S} \boldsymbol{P}_{i,j} > 0$.*

**Assumption 3.2.** *The action observation matrix $\boldsymbol{O} \in \mathbb{R}^{IV \times S}$ is full column rank.*

Basically, the first assumption gives a non-null probability of transitioning from any state to any other. This assumption implies the regularity of the chain and, consequently, the presence of a unique stationary distribution, as shown in Proposition 3.1. Furthermore, it also ensures geometric ergodicity as defined in Krishnamurthy (2016), which is a condition that allows the Chain to reach its stationary distribution $\boldsymbol{\pi}$ geometrically fast, regardless of its initial distribution. Our assumption on the minimum entry is not a necessary condition for the two aforementioned motivations but a sufficient one. However, we require this condition in order to bound the error in the belief computed using an estimated transition matrix and the real one. This result is presented in Proposition C.4 and builds on a result from De Castro et al. (2017). The same assumption is also used in other works dealing with non-episodic interactions such as Zhou et al. (2021); Mattila et al. (2020), while this assumption is not necessary in works such as Azizzadenesheli et al. (2016) since the used policies are memoryless, thus not making use of a belief state.

The second assumption, instead, is related to the identifiability of the parameters of the problem and has been largely used in works using spectral decomposition techniques (Zhou et al., 2021; Azizzadenesheli et al., 2016; Hsu et al., 2012) and other works involving learning of parameters in POMDPs (Liu et al., 2022; Jin et al., 2020). In the following, we will see that this is a necessary condition in order to recover the matrix $\boldsymbol{P}$. Indeed, we will see that the estimation procedure scales with a term $\sigma_{\min}^2(\boldsymbol{O})$ at the denominator and through Assumption 3.2, we require that $\sigma_{\min}(\boldsymbol{O}) > 0$.

## 4 Proposed Approach

### 4.1 Markov Chain Estimation

As previously stated, the objective is to learn the transition matrix $\boldsymbol{P}$ using the observations we get from the different pulled arms assuming to know the tensor $\boldsymbol{\mathsf{O}} \in \mathbb{R}^{S \times I \times V}$. First of all, we start with a consideration about the transition matrix that defines the chain. Building on Assumption 3.1 and following Proposition 3.1, we can say that exists a unique stationary distribution. This distribution can be easily found by solving the equation below:

$$\boldsymbol{\pi} \boldsymbol{P} = \boldsymbol{\pi}.$$

From the stationary distribution $\boldsymbol{\pi}$, we can define the diagonal matrix $\boldsymbol{\Pi} = diag(\boldsymbol{\pi})$ having the values of the stationary distribution along its diagonal, and we can define the matrix $\boldsymbol{W} = \boldsymbol{\Pi} \boldsymbol{P}$ satisfying $\sum_{i,j \in S} W_{i,j} = 1$. We can see matrix $\boldsymbol{W}$ as the transition matrix $\boldsymbol{P}$ where the transition probabilities from each state (reported along the rows of the transition matrix) are scaled by the probability of the state, given by the stationary distribution. Having defined the matrix $\boldsymbol{W}$, we can interpret the element $W_{i,j}$ as the probability of seeing the transition from state $s_i$ to state $s_j$ when the two consecutive pairs of states are sampled from the mixed Chain. We will also refer to $\boldsymbol{W}$ as the stationary transition distribution matrix. Our objective will be to build an estimate $\widehat{\boldsymbol{W}}$ of the $\boldsymbol{W}$ matrix from which we will derive $\widehat{\boldsymbol{P}}$.

Let's now define an exploration policy $\theta$ that selects pairs of arms to be played in successive rounds. We use this policy for $T_0$ episodes on MABs that switch according to the underlying Markov Chain, and we obtain a sequence $\mathbb{D} = \{(a_1, r_1), (a_2, r_2), \ldots, (a_{T_0}, r_{T_0})\}$. This sequence can also be represented by combining non-overlapping pairs of consecutive elements, thus obtaining $Pairs(\mathbb{D}) = \{(a_1, a_2, r_1, r_2), \ldots, (a_{T_0-1}, a_{T_0}, r_{T_0-1}, r_{T_0})\}$.

We introduce now the vector $\mathbf{n}_{T_0} \in \mathbb{N}^{I^2 V^2}$ that counts the number of occurrences of elements in $Pairs(\mathbb{D})$. More formally, for each cell of the vector $\mathbf{n}_{T_0}$, we have:

$$\mathbf{n}_{T_0}(a_i, a_j, r_i, r_j) = \sum_{t=0}^{T_0/2} \mathbb{1}\{I_{2t} = a_i, I_{2t+1} = a_j, R_{2t} = r_i, R_{2t+1} = r_j\}.$$

Given the previous considerations, we are now ready to state a core result that links the stationary transition distribution matrix $\boldsymbol{W}$ and the count vector $\mathbf{n}_{T_0}$ as follows:

$$\mathbb{E}[\mathbf{n}_{T_0}(a_i, a_j, r_i, r_j)] = \sum_{s_i, s_j} W_{s_i, s_j} \sum_{t=0}^{T_0/2} \theta(I_{2t} = a_i, I_{2t+1} = a_j) P((R_{2t} = r_i, R_{2t+1} = r_j)|(a_i, a_j), (s_i, s_j)). \quad (7)$$

This equation basically states that the expected value of an element of the count vector $\mathbb{E}[\mathbf{n}_{T_0}(a_i, a_j, r_i, r_j)]$ depends on the probability of sampling the corresponding couple of arms and on the conditional probabilities of rewards given the couple of arms and the couple of states, with each of this probability being weighted by the probability $W_{s_i, s_j}$ that each state transition occurs. We can write the previous formulation in matrix form as follows:

$$\mathbb{E}[\mathbf{n}_{T_0}] = \frac{T_0}{2} \boldsymbol{D} \boldsymbol{A} \boldsymbol{w}, \quad (8)$$

where the matrix $\boldsymbol{A}$ is the reference matrix already defined in Section 3.3, vector $\boldsymbol{w} = Vec(\boldsymbol{W})$ is the vectorization of the matrix $\boldsymbol{W}$, while $\boldsymbol{D} \in \mathbb{R}^{I^2 V^2}$ is a diagonal matrix containing the probabilities (determined by policy $\theta$) associated to each combination of arms, each appearing with multiplicity $V^2$.

Having defined Equation 8, we are able to compute an estimate of the vector $\widehat{\boldsymbol{w}}$ based on the obtained vector count $\mathbf{n}_{T_0}$:

$$\widehat{\boldsymbol{w}} = \boldsymbol{A}^\dagger \widehat{\boldsymbol{D}}_{T_0}^{-1} \mathbf{n}_{T_0}, \quad (9)$$

where $\boldsymbol{A}^\dagger$ is the Moore–Penrose inverse of reference matrix $\boldsymbol{A}$ and matrix $\widehat{\boldsymbol{D}}_{T_0}$ is the diagonal matrix that counts with multiplicity $V^2$ the number of occurrences of each combination of arms (we assume that each

combination of arms has been pulled at least once, so $\widehat{\boldsymbol{D}}_{T_0}$ is invertible).

In the limit of infinite samples, Equation 9 has a fixed exact solution that is $\widehat{\boldsymbol{w}} = \boldsymbol{w}$.

After the computation of $\widehat{\boldsymbol{w}}$, we obtain an estimate of $\widehat{\boldsymbol{P}}$. The derivation from $\widehat{\boldsymbol{w}}$ to $\widehat{\boldsymbol{P}}$ implies two main steps: the first is to write back the vector $\widehat{\boldsymbol{w}}$ in matrix form, reversing the vectorization operation and obtaining matrix $\widehat{\boldsymbol{W}}$; the second step consists in normalizing each obtained row so that the values on each row sum to 1, thus deriving the stochastic matrix $\widehat{\boldsymbol{P}}$.

## 4.2 SL-EC Algorithm

Having established an estimation procedure for the transition matrix $\widehat{\boldsymbol{P}}$, we will now provide an algorithm that makes use of this approach in a regret minimization framework.

We consider a finite horizon $T$ for our problem. We propose an algorithm called *Switching Latent Explore then Commit* (SL-EC) that proceeds using an EC approach where the exploration phase is devoted to finding the best estimation of the transition matrix $\widehat{\boldsymbol{P}}$, while during the exploitation phase, we maximize the instantaneous expected reward using the information contained in the belief state $\mathbf{b}$ with the formulation provided in Equation 6. The Exploration phase lasts for $T_0$ episodes, where $T_0$ is optimized w.r.t. the total horizon $T$, as will be seen in Equation 15.

The presented approach is explained in the pseudocode of Algorithm 1.

Basically, a set of all the ordered combinations of pairs of arms is generated at the beginning of the exploration phase, and the pairs of arms are sequentially pulled in a round-robin fashion until the exploration phase is over. The choice of a round-robin approach allows the highlighting of some interesting properties in the theoretical analysis, as will be shown later in Section 5. When the exploration phase is over, an estimation of the transition matrix $\widehat{\boldsymbol{P}}$ is computed using the procedure described in Section 4.1. After that, a belief vector $\mathbf{b}$ is initialized, assigning a uniform probability to all the states, and it is updated using the estimated $\widehat{\boldsymbol{P}}$, considering the history of samples collected during the exploration phase up to $T_0$. Finally, the exploitation phase starts, as described in the pseudocode of the algorithm.

## 4.3 Arm Selection Policy

In Algorithm 1, we propose a simple approach for choosing the arms to pull. Each ordered combination of pairs of arms is indeed pulled the same number of times during the exploration phase by using a deterministic approach. However, the estimation framework proposed in Section 4.1 allows for a more flexible arm selection policy. We may randomize the arm choice by assigning non-uniform probabilities to each combination. This aspect allows exploiting the knowledge of the known reward distribution of each arm, for example, giving a higher probability to the combinations of arms that are more rewarding (assuming an initial uniform distribution over state transitions). This arm selection policy may be particularly efficient if we plug this estimation framework into an iterative two-phase exploration and exploitation algorithm, as that used in Zhou et al. (2021). Notably, we could use the estimates of the transition matrix $\widehat{\boldsymbol{P}}_k$ at the end of the $k$-th exploration phase to properly modify the exploration policy in phase $k+1$ by giving higher probabilities to combinations of arms that are expected to be more rewarding. Indeed, our approach is able to reuse all samples collected during previous exploration phases despite being drawn using different exploration policies.

**Offline arm selection** In problems with a large number of available arms, a round-robin approach among all possible combinations of pairs may be detrimental as it considers all arms equivalently. There may be cases where some actions carry less information. The extreme case is an action that induces the same observation distribution for all the switching MABs. Indeed, pulling that action will not provide any additional information on the current MAB and the effect will only be to slow down the estimation procedure. In general, actions that induce *similar* observation distributions for all the MABs will provide *less* information with respect to actions that induce highly different distributions for all the MABs.

A more convenient approach, in this case, would be to select a subset of different arms, thus leading to a limited number of combinations of pairs of arms to use during the exploration phase. Clearly, in the general case, the removal of some arms may lead to a loss of the total information available. Intuitively, the arm selection procedure tends to promote diversity among arms given the latent states, in order to increase the identifiability capabilities deriving from the actions. It turns out that we are able to get an understanding

---

**Algorithm 1:** SL-EC Algorithm

**Input:** Reference Matrix $\boldsymbol{A}$, Exploration horizon $T_0$, Total horizon $T$

**1** Initialize vector of counts $\mathbf{n} \in \mathbb{N}^{I^2 V^2}$ with zeroes
**2** $t \leftarrow 0$
**3** $\mathbb{D} \leftarrow \{\}$
**4** **while** $t \leq T_0$ **do**
**5**      **foreach** $(a_i, a_j) \in I^2$ **do**
**6**          Pull arm $I_t = a_i$
**7**          Observe reward $\mathbf{r}_t$
**8**          Pull arm $I_{t+1} = a_j$
**9**          Observe reward $\mathbf{r}_{t+1}$
**10**          Update $\mathbf{n}$ with $(I_t, I_{t+1}, \mathbf{r}_t, \mathbf{r}_{t+1})$
**11**          $\mathbb{D}.add((I_t, \mathbf{r}_t), (I_{t+1}, , \mathbf{r}_{t+1}))$
**12**          $t \leftarrow t + 2$

**13** $\widehat{\boldsymbol{w}} \leftarrow$ Use Equation 9
**14** $\widehat{\boldsymbol{P}} \leftarrow$ Compute Transition Matrix$(\widehat{\boldsymbol{w}})$
**15** $t \leftarrow 0$
**16** $\mathbf{b}_0 \leftarrow Uniform()$
**17** **while** $t \leq T$ **do**
**18**      **if** $t \leq T_0$ **then**
**19**          $I_t = \mathbb{D}.getAction(t)$
**20**      **else**
**21**          $I_t = \arg\max_{a \in \mathbb{I}} \sum_{s \in \mathbb{S}} \sum_{r \in \mathbb{V}} rQ(r|s, a)\mathbf{b}_{s,t}$
**22**      Observe reward $\mathbf{r}_t$
**23**      $\mathbf{b}_{t+1} \leftarrow UpdateBelief(\mathbf{b}_t, I_t, \mathbf{r}_t)$
**24**      $t \leftarrow t + 1$

---

of the information loss we suffer by selecting specific arms, given the knowledge of the action observation matrix $\boldsymbol{O}$. In particular, in Section 5 devoted to the theoretical analysis, we will see that the expression $\frac{1}{\sigma_{\min}^2(\boldsymbol{O})}$, with $\sigma_{\min}(\boldsymbol{O})$ representing the minimum singular value of the action observation matrix $\boldsymbol{O}$, is an index of the complexity of the estimation procedure and we can use this value to drive the choice of the best subset of arms to use.

In particular, by fixing a number $J < I$ of arms to use among those available, the choice over the best subset of size $J$ can be done as follows. We consider all the possible subsets of arms of size $J$ and for each of these subsets, we derive a reduced action observation matrix $\boldsymbol{G}$ of size $JV \times S$ that is obtained by simply removing from the original matrix $\boldsymbol{O}$ all the rows associated to the actions not belonging to the considered subset of arms. Intuitively, for each generated subset, this procedure corresponds to redefining new simplified MAB instances having as actions only those appearing in the subset. From these reduced action observation matrices, it is possible to derive new reduced reference matrices, as described in Section 3.3. Having defined a new action observation matrix for each generated subset, their minimum singular values are compared, and a good candidate subset of arms is the one yielding the reduced action observation matrix $\boldsymbol{G}$ with the highest $\sigma_{\min}(\boldsymbol{G})$. Understandably, this approach implies that the reduced action observation matrix $\boldsymbol{G}$ derived from the subset of selected arms should be full-column rank, thus satisfying Assumption 3.2. It follows that the necessary condition $JV \geq S$ should be verified.

## 5 Theoretical Analysis

We will now provide theoretical guarantees on the matrix estimation procedure presented in Section 4.1 and we will prove a regret bound for the SL-EC Algorithm.

We start with a concentration bound on the transition matrix $\widehat{\boldsymbol{P}}$ estimated using samples coming from a round-robin collection policy.

**Lemma 5.1.** *Suppose Assumption 3.2 holds and suppose that the Markov chain with transition matrix $\boldsymbol{P}$ starts from its stationary distribution $\boldsymbol{\pi} \in \Delta^{S-1}$ and that $\pi_{\min} := \min_s \boldsymbol{\pi}(s) > 0$. By fixing an exploration parameter $T_0$ and by pulling each combination of pairs of arms in a round-robin fashion, with probability at least $1 - \delta$ the estimation error of the transition matrix $\boldsymbol{P}$ will be:*

$$\|\boldsymbol{P} - \widehat{\boldsymbol{P}}\|_F \leq \frac{4I^2}{\sigma_{\min}^2(\boldsymbol{O})\pi_{\min}} \sqrt{\frac{S(1 + \log(I^2/\delta))}{(1 - \lambda^{2I^2})T_0}}, \tag{10}$$

where $\|\cdot\|_F$ represents the Frobenius norm (Golub & Van Loan, 1996), $\sigma_{\min}$ represents the minimum singular value of the reference matrix $\boldsymbol{A}$, and $\lambda$ represents the second highest eigenvalue of matrix $\boldsymbol{P}$. We will provide here a sketch of the proof of the presented Lemma. A more detailed version of this proof is reported in Appendix B.

**Sketch of the proof**   The proof of Lemma 5.1 builds on two principal results. The former comprises a relation that links the estimation error of the matrix $\boldsymbol{P}$ with the estimation error of the stationary transition distribution matrix $\boldsymbol{W}$, while the latter is a concentration bound on the estimated $\widehat{\boldsymbol{W}}$ from the true one $\boldsymbol{W}$. Concerning the first result, we can say that:

$$\|\boldsymbol{P} - \widehat{\boldsymbol{P}}\|_F \leq \frac{2\sqrt{S}\|\boldsymbol{W} - \widehat{\boldsymbol{W}}\|_F}{\pi_{\min}}. \tag{11}$$

This result follows from a sequence of algebraic manipulations, also involving a derivation from (Ramponi et al., 2020).

We now need to define a bound on $\|\boldsymbol{W} - \widehat{\boldsymbol{W}}\|_F$. In order to bound this quantity, we apply the vectorization operator $Vec(\cdot)$ to the two matrices obtaining respectively $\boldsymbol{w}$ and $\widehat{\boldsymbol{w}}$ and use the fact that $\|\boldsymbol{W} - \widehat{\boldsymbol{W}}\|_F = \|\boldsymbol{w} - \widehat{\boldsymbol{w}}\|_2$. We proceed as follows:

$$\|\boldsymbol{w} - \widehat{\boldsymbol{w}}_{T_0}\|_2 = \left\| \frac{2}{T_0} A^\dagger \boldsymbol{D}^{-1}(\mathbb{E}[\mathbf{n}_{T_0}] - \mathbf{n}_{T_0}) \right\|_2 = \left\| A^\dagger (\mathbf{z} - \widehat{\mathbf{z}}) \right\|_2$$

$$\leq \|\boldsymbol{A}^\dagger\|_2 \|\mathbf{z} - \widehat{\mathbf{z}}\|_2 = \frac{1}{\sigma_{\min}(\boldsymbol{A})}\|\mathbf{z} - \widehat{\mathbf{z}}\|_2 = \frac{1}{\sigma_{\min}^2(\boldsymbol{O})}\|\mathbf{z} - \widehat{\mathbf{z}}\|_2, \tag{12}$$

where in the second equality we have introduced the vector $\mathbf{z} := (2/T_0)\boldsymbol{D}^{-1}\mathbb{E}[\mathbf{n}_{T_0}] \in \mathbb{R}^{I^2 V^2}$ and its estimate $\widehat{\mathbf{z}}$. Basically, vector $\widehat{\mathbf{z}}$ is obtained by dividing each component of the vector count $\mathbf{n}_{T_0}$ by the number of times the combination of arms associated with that component has been pulled. In the inequality instead, we used the consistency property for the spectral norm of matrix $\boldsymbol{A}^\dagger$, while in the last equality we used the result in Equation 4.

Given the vector $\widehat{\mathbf{z}}$, we can see it as a concatenation of $I^2$ vectors $\widehat{\mathbf{m}}_k \in \mathbb{R}^{V^2}$, with $k \in \{1, \ldots, I^2\}$ where each vector $\widehat{\mathbf{m}}_k$ is the empirical estimate of vector $\mathbf{m}_k$, and it is obtained from $\frac{T_0}{2I^2}$ samples. Vector $\mathbf{m}_k$ represents the parameters of a discrete distribution over the pair of observations conditioned on the pair of actions $k$. By definition, we have:

$$\|\widehat{\mathbf{z}} - \mathbf{z}\|_2 = \sqrt{\sum_{k=1}^{I^2} \|\widehat{\mathbf{m}}_k - \mathbf{m}_k\|_2^2}. \tag{13}$$

We are now able to bound the error in the estimated distribution $\mathbf{m}_k$ by using a result shown in Proposition C.2 and is inspired by the work of Hsu et al. (2012). With probability at least $1 - \delta/I^2$ we have that:

$$\|\widehat{\mathbf{m}}_k - \mathbf{m}_k\|_2 \leq \sqrt{\left(\frac{1 + \lambda^{2I^2}}{1 - \lambda^{2I^2}}\right) \frac{2I^2(1 + \log(I^2/\delta))}{T_0}}.$$

The exponential term $2I^2$ that appears to the second highest eigenvalue $\lambda$ has been introduced thanks to the adoption of the round-robin procedure for the choice of combinations of arms. Notably, each combination is pulled every $2I^2$ steps of the Markov Process, thus resulting in a faster mixing of the chain. For more details, please refer to Appendix B.

By combining the last obtained bound with 12 and 13 and using a union bound on all the vectors of type $\mathbf{m}_k$, we have that with probability at least $1 - \delta$:

$$\|\boldsymbol{w} - \widehat{\boldsymbol{w}}_{T_0}\|_2 \leq \frac{1}{\sigma_{\min}^2(\boldsymbol{O})} \sqrt{\left(\frac{1 + \lambda^{2I^2}}{1 - \lambda^{2I^2}}\right) \frac{2I^4(1 + \log(I^2/\delta))}{T_0}}$$

$$\leq \frac{2I^2}{\sigma_{\min}^2(\boldsymbol{O})} \sqrt{\frac{1 + \log(I^2/\delta)}{(1 - \lambda^{2I^2})T_0}}.$$

Ultimately, by putting together the bound just obtained with 11, we are able to obtain the final result stated in the Lemma.

Having established the results on the estimation matrix $\boldsymbol{P}$, we can now provide regret guarantees for Algorithm 1. The oracle we use is aware of both the observation tensor $\mathbf{O}$ and the transition matrix $\boldsymbol{P}$ but does not observe the hidden state. As well as our algorithm, it builds a belief over the states, using the formulation defined in Equation 5 and selects the arm maximizing the expected instantaneous reward. The derived regret upper bound is provided in the following:

**Theorem 5.1.** *Suppose Assumptions 3.1 and 3.2 hold and suppose that the Markov chain with transition matrix $\boldsymbol{P}$ starts from its stationary distribution $\boldsymbol{\pi} \in \Delta^{S-1}$ and that $\pi_{\min} := \min_s \boldsymbol{\pi}(s) > 0$. By considering a finite horizon $T$, there exists a constant $T_0$, with $T > T_0$, such that with probability at least $1 - \delta$, the regret of the SL-EC Algorithm satisfies:*

$$\mathfrak{R}(T) \leq 2 \left( \frac{2LI^2}{\sigma_{\min}^2(\boldsymbol{O})\pi_{\min}} \sqrt{\frac{S(1 + \log(I^2/\delta))}{(1 - \lambda^{2I^2})}} \cdot T \right)^{2/3}, \tag{14}$$

where $L = \frac{4S(1-\epsilon)^2}{\epsilon^3} + \sqrt{S}$ is a constant that is used to bound the error in the estimated belief (more details in Proposition C.4 in the Appendix). The presented regret has an order of $\mathcal{O}(T^{2/3})$ w.r.t the horizon $T$, as common when using an Explore-Then-Commit algorithm. A detailed proof of this theorem can be found in Appendix B. The presented bound on the regret can be achieved by appropriately choosing the exploration horizon $T_0$. More specifically, we set it as follows:

$$T_0 = \left( \frac{2LTI^2}{\sigma_{\min}^2(\boldsymbol{O})\pi_{\min}} \sqrt{\frac{S(1 + \log(I^2/\delta))}{(1 - \lambda^{2I^2})}} \right)^{2/3}. \tag{15}$$

## 5.1 Dependency on the Problem Parameters

By analyzing the results on the bound of the regret, we can observe that it scales with $I^2$. This may seem concerning especially when dealing with problems involving a high number of arms. Furthermore, the analyzed problem is presented for discrete observations and we are not able to handle this configuration in the case with continuous reward models as the number of observations would be infinite, hence impeding the construction of the reference matrix. Fortunately, we can address both aspects, the one related to the dependency on the number of arms and the other on the dependency on the number of observations.

**Dependency on the Number of Arms** Considering the number of arms, we already observed in Section 4.3 that when the number of arms is large, it is possible to select a subset of arms that allows solving the problem. In particular, the best subset $\mathbb{J}$ we can select is the one minimizing the term $\frac{J^2}{\sigma_{\min}^2(\boldsymbol{G}_{\mathbb{J}})}$, with $J$ being the size of $\mathbb{J}$ and $\boldsymbol{G}_{\mathbb{J}}$ being the reduced action observation matrix obtained from the choice of the arms in $\mathbb{J}$. It is indeed likely that when $I \gg S$, some arms contain redundant information and can be easily discarded for the estimation procedure.

**Continuous Reward Distributions**  Concerning the number of observations, it appears that handling continuous reward distributions within this framework is not feasible and this is true if we apply our approach as is. However, nothing prevents us from discretizing the observation distributions and considering the discretized distribution as a categorical one. The process of discretization involves dividing the continuous observation distributions into a predetermined number $U$ of distinct consecutive intervals. Each interval is assigned a probability value that represents the likelihood of a particular sample originating from the continuous distribution and belonging to that interval. Formally, if we consider having a Switching Latent Bandit problem with continuous rewards and a number $S$ of bandits and a number $I$ of actions available for each bandit, there will be $j \in \{1, 2, \ldots, SI\}$ potentially different continuous reward distributions $P_j(r)$ where $j$ identifies a specific state-action pair. If we assume to discretize each reward distribution into $U$ consecutive intervals, we will have $U-1$ splitting points. If we consider now the ordered set of splitting points and take two consecutive splitting points $u_h$ and $u_k$ for which holds that $u_h < u_k$, we can define the interval $\mathcal{I}_{hk} = (u_h, u_k]$. The probability that a realization from a continuous distribution $P_j$ falls within interval $\mathcal{I}_{hk}$ is defined as $P_j(r \in \mathcal{I}_{hk}) = \int_{u_h}^{u_k} P_j(r) dr$. Of course, if we are able to exactly compute the integrals in the previous formulation we will not introduce any error in the discretization process. By applying the same procedure for all the $U$ intervals identified, we can define the parameters of the new categorical distribution. This procedure is then iterated for all the continuous probabilities $P_j$ using the same splitting points and we finally obtain a new action observation matrix of size $IU \times S$, which should of course satisfy Assumption 3.2. From this point on, we can build the new reference matrix and we can proceed with the SL-EC algorithm.

 It is an interesting problem to determine in this setting the number of suitable splits and the location of the split points that leads to a higher $\sigma_{\min}(\boldsymbol{O})$.

Another issue arises when the environment comprises numerous but finite observations. In such scenarios, we can employ the inverse approach by clustering some observations, thereby reducing the scale of the problem. By selecting a number of clusters $C < V$, we can divide the observations into distinct groups. This allows us to utilize cluster-level probabilities (obtained by summing probabilities of the single observations) to construct a new action observation matrix.

## 5.2  Comparison with the SEEU algorithm (Zhou et al., 2021)

We devote this section to the comparison of our work with that of Zhou et al. (2021) and with the general spectral decomposition techniques (Anandkumar et al., 2014). We start by highlighting the main differences with the cited work. In particular:

- they consider learning both the transition and the observation models, while we assume to know the latter.

- they have a further assumption compared to ours as they require the invertibility of the transition matrix $\boldsymbol{P}$.

- they assume to have access to an optimization oracle that returns the optimal policy for a given known model. Differently, our oracle optimizes the best instantaneous expected reward given the belief on the MABs at each timestep computed using the real transition and observation matrices.

- as a minor difference, they explicitly focus on the case with Bernoulli observations, even if their results can be extended, as happens in our case, also to a general number of discrete observations.

The authors propose the SEEU (Spectral Exploration and Exploitation with UCB) algorithm that alternates between exploration phases used to make parameter estimation and exploitation phases where the actions are pulled according to the computed optimistic policy. During the exploration phase, they estimate both the observation and the transition models through techniques based on standard spectral decomposition methods (Anandkumar et al., 2014). The guarantees they provide hold under both our Assumption 3.1 and 3.2 and they further require the invertibility of the transition matrix. The devised algorithm reaches $\mathcal{O}(T^{2/3})$ regret, disregarding logarithmic terms.

**Comparison with Spectral Decomposition Techniques**    First of all, we need to introduce some quantities that will be helpful in what will follow. We will report some results appearing in Appendix B of Zhou et al. (2021) on spectral decomposition techniques, based on the work of Anandkumar et al. (2014).

We assume to encode the couple action-reward into a new variable $q \in \{1, 2, \ldots, IV\}$ through a one-to-one mapping. The observable random vector $I_t, R_t$ is rewritten as a random variable $Q_t$. Hence, it is possible to define the following matrices $\boldsymbol{B}_1, \boldsymbol{B}_2, \boldsymbol{B}_3 \in \mathbb{R}^{IV \times S}$ as follows:

$$\boldsymbol{B}_1(q, s) = P(Q_{t-1} = q | S_t = s)$$
$$\boldsymbol{B}_2(q, s) = P(Q_t = q | S_t = s)$$
$$\boldsymbol{B}_3(q, s) = P(Q_{t+1} = q | S_t = s)$$

for $q \in \{1, 2, \ldots, IV\}$ and $s \in \{1, \ldots, S\}$. It is now important to note the similarities between matrix $\boldsymbol{B}_2$ and our action observation matrix $\boldsymbol{O}$. Given any state $s \in \mathbb{S}$ and any variable $q$ that maps the pair $(a, r)$, we have that:

$$\boldsymbol{O}(q, s)P(I_t = a | S_t = s) = Q(R_t = r | S_t = s, I_t = a)P(I_t = a | S_t = s)$$
$$= P(R_t = r, I_t = a | S_t = s)$$
$$= P(Q_t = q | S_t = s) = \boldsymbol{B}_2(q, s),$$

where the first equality follows by the definition of $\boldsymbol{O}$ in Equation 2. Furthermore, since the SEEU algorithm samples uniformly over the $I$ actions during the exploration phase, for any $q \in \{1, 2, \ldots, IV\}$ and $s \in \mathbb{S}$, we have that:

$$\boldsymbol{O}(q, s)P(I_t = a | S_t = s) = \boldsymbol{O}(q, s)\frac{1}{I} = \boldsymbol{B}_2(q, s).$$

Thus, the stated result allows also to say:

$$\sigma_{\min}(\boldsymbol{O}) = \sigma_{\min}(\boldsymbol{B}_2)I. \tag{16}$$

We can now present the bound on the error of the estimated transition matrix. From Anandkumar et al. (2014), it can be shown that with a sufficient number of samples $N$, with probability at least $1 - \delta$, it holds that:

$$\|\boldsymbol{P} - \widehat{\boldsymbol{P}}\|_F \leq C_2 \sqrt{\frac{\log\left(6\frac{I^2V^2 + IV}{\delta}\right)}{N}}, \tag{17}$$

with

$$C_2 = \frac{4}{\sigma_{\min}(\boldsymbol{B}_2)}\left(S + S^{3/2} * \frac{21}{\sigma_{1,-1}}\right)C_3$$

$$C_3 = 2G\frac{2\sqrt{2} + 1}{(1 - \theta)\sqrt{\pi_{\min}}}\left(1 + \frac{8\sqrt{2}}{\pi_{\min}^2\sigma^3} + \frac{256}{\pi_{\min}^3\sigma^2}\right)$$

where $\sigma_{1,-1}$ is the smallest nonzero singular value of a covariance matrix computed during the estimation process (see Section 3.1 in Zhou et al. (2021)) and $\sigma = \min\{\sigma_{\min}(\boldsymbol{B}_1), \sigma_{\min}(\boldsymbol{B}_2), \sigma_{\min}(\boldsymbol{B}_3)\}$, where $\sigma_{\min}(\boldsymbol{B}_i)$ represents the smallest nonzero singular value of the matrix $\boldsymbol{B}_i$, for $i = 1, 2, 3$.

*bmpi* represents the stationary distribution of the underlying Chain and $\pi_{\min} := \min_s \boldsymbol{\pi}(s) \geq \epsilon$. Finally, $\theta$ and $G$ are some mixing rate parameters. Under Assumption 3.1, we can take $G = 2$ and have that $\theta \leq 1 - \epsilon$. In Equation 17, we have reported the bound with respect to the Frobenius norm in order to be aligned with the result shown in Lemma 5.1. This is different from what is reported in Zhou et al. (2021) where the bound is with respect to the spectral norm. To make the conversion, we used the fact that $\|\boldsymbol{P} - \widehat{\boldsymbol{P}}\|_F \leq \sqrt{S}\|\boldsymbol{P} - \widehat{\boldsymbol{P}}\|_2$ and inserted a further $\sqrt{S}$ in the definition of $C_2$.

We can simplify the expression of $C_2$ by considering a different expression such that:

$$C_2' = \frac{16(2\sqrt{2} + 1)}{\sigma_{\min}(\boldsymbol{B}_2)\sqrt{\pi_{\min}}}\left(S + S^{3/2} * \frac{21}{\sigma_{1,-1}}\right)\left(1 + \frac{8\sqrt{2}}{\pi_{\min}^2\sigma_{\min}(\boldsymbol{B}_2)^3} + \frac{256}{\pi_{\min}^3\sigma_{\min}(\boldsymbol{B}_2)^2}\right),$$

resulting that $C_2' \leq C_2$. We recall now the equivalent term that bounds the error in the estimated matrix appearing in our Lemma 5.1 that is:

$$C_{our} = \frac{4I^2\sqrt{S}}{\sigma_{\min}^2(\boldsymbol{O})\pi_{\min}} = \frac{4\sqrt{S}}{\sigma_{\min}^2(\boldsymbol{B}_2)\pi_{\min}}$$

where the first equality follows from Equation 16.

By comparing these quantities, we can see that the dependency $C_2'$ has on the different parameters of the problem is generally worse than that of $C_{our}$. $C_2'$ has a dependency of order $-7/2$ with respect to $\pi_{\min}$ while $C_{our}$ enjoys a dependency of order $-1$. By considering instead the number of states $S$, their constant contains a term that scales with order $3/2$, while we have a dependency of order $1/2$. Finally, the dependency on the minimum singular value of the action observation matrix has order $-4$ in $C_2'$ and order $-2$ in $C_{our}$. Finally, we have an explicit squared dependency on the number of actions, while this dependency is hidden in the definition of the different $\boldsymbol{B}_i$ matrices in the SD techniques. Again, we recall that these considerations are made on $C_2'$ which is a smaller value than the real one $C_2$ appearing in their bound.

**Comparison with the Regret of the SEEU Algorithm** After having assessed the differences in the estimation procedure of the transition matrix $\boldsymbol{P}$, we analyze the difference in the regret of the two algorithms. The terms appearing in the regret of SEEU are quite involved as they contain quantities that are of course related to the choice of the oracle and to some algorithm-specific quantities. By rewriting the expression of the regret of the SEEU algorithm, we have $\mathfrak{R}_T \leq CT^{2/3}$, with $C$ defined as:

$$C = \frac{D\tau_2^{1/3}}{2\tau_1^{1/2}}\Big(L_1 S^{3/2}C_1 + L_2 S^{1/2}C_2\Big) + F \tag{18}$$

where F contains other problem-dependent parameters that are irrelevant for our comparison, the term $D$ is a uniform bound on the span of the bias term appearing in the Bellman Equation they use for their policy, while $\tau_1$ and $\tau_2$ are related to the duration of the exploration and exploitation phases. We are interested in the remaining terms appearing inside the parentheses and we will focus in particular on the second term $L_2 S^{1/2}C_2$ which depends on the quantity $C_2$ we have just analyzed.

The regret of our SL-EC algorithm can be bounded by a term that is $(LC_{our})^{2/3}$ where the quantity $L$ corresponds exactly to quantity $L_2$. By only considering the term $L_2 S^{1/2}C_2$ for the comparison, we see that this term contains a further $\sqrt{S}$ dependency on the number of states. Finally, we stress the fact that the constants appearing in our bound are all scaled with an order of $2/3$, differently from the constants appearing in the regret of the SEEU algorithm.

## 6 Numerical Simulations

In this section, we provide numerical simulations on synthetic and semi-synthetic data based on the Movie-Lens 1M (Harper & Konstan, 2015) dataset, demonstrating the effectiveness of the proposed Markov Chain estimation procedure. Specifically, we show the efficiency of the offline arm selection procedure described in Section 4.3 and conduct a comparison between our SL-EC Algorithm and several baselines in non-stationary settings. In Section 6.3, we provide additional experiments that highlight the performance difference between our approach and a modified technique based on Spectral Decomposition methods.

### 6.1 Estimation Error of Transition Matrix

The first set of experiments is devoted to showing the error incurred by the estimation procedure of the transition matrix in relation to the number of samples considered and the set of actions used for estimation. The left side of Figure 1 illustrates the estimation error of the transition matrix given different instances of Switching Bandits with an increasing number of states. In particular, we fix the number of total actions $I = 10$ and number of observations $V = 10$ and consider three instances with $S = 5$, $S = 10$ and $S = 15$ number of states. As it is expected, we can see that as the number of states increases the problem becomes more complex, and more samples are needed in order to improve the estimation. Figure 1 reports the

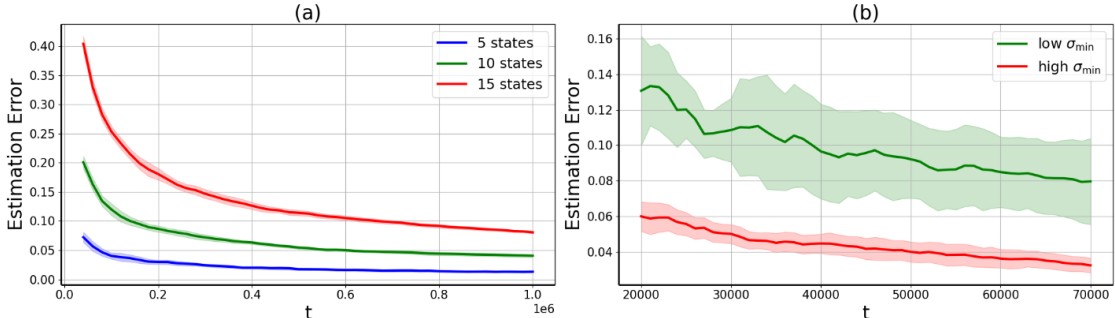

Figure 1: (a) Difference between the estimated and real transition matrix with an increasing number of samples. The metric used is $\|\cdot\|_F$ (10 runs, 95% c.i.), (b) Difference between real and estimated transition matrix using two different subsets of arms of size $J = 3$ arms from the 8 available on a problem with 5 states. The metric used is $\|\cdot\|_F$ (10 runs, 95% c.i.).

Frobenius norm $\|\cdot\|_F$ of the error between the true and the estimated transition matrix. We can see that the estimation procedure is particularly efficient leading to low error values even with a limited number of samples, as can be seen from the steep error drop experienced in the first part of the plot.

The right plot in Figure 1, instead, shows the estimation error obtained by using a different subset of arms. As mentioned in previous sections, it is not always beneficial to use all the available actions during the estimation procedure, but selecting a subset of actions may be preferable. Furthermore, we show that by selecting specific subsets of arms we can improve the estimation w.r.t using other subsets. For this experiment, we consider $J = 3$ arms among the $I = 8$ available for a Switching MAB instance with $S = 5$ states. We then identify the optimal subset of arms of size $J$ and initiate the estimation process using the selected subset. In order to find the best one, we generate all matrices of type $\boldsymbol{G}$, as described in Section 4.3 and choose the matrix with the highest $\sigma_{\min}(\boldsymbol{G})$. The subset of arms generating that matrix will be used for estimation. The estimation error of the best subset of arms is represented in the plot with the red line, while we represent in green the estimation error of the subset having the lowest $\sigma_{\min}(\boldsymbol{G})$. The figure clearly exhibits the performance difference between the two choices, thereby validating our claims. Additional details about the characteristics of the matrices used in the experiments are provided in Appendix A.

## 6.2 Algorithms Comparisons

In this second set of experiments, we compare the regret suffered by our SL-EC approach with other algorithms specifically designed for non-stationary environments. Following the recent work of Zhou et al. (2021), we consider the subsequent baseline algorithms: the simple $\epsilon$-*greedy* heuristics, a sliding-window algorithm such as *SW-UCB* (Garivier & Moulines, 2011) that is generally able to deal with non-stationary settings and the *Exp3.S* (Auer et al., 2002) algorithm. The parameters for all the baseline algorithms have been properly tuned according to the different considered settings. It is worth noting that, unlike our Algorithm, the baseline algorithms do not have knowledge of the observation tensor or the underlying Markov Chain. In contrast, our approach utilizes the observation tensor to estimate the transition matrix and to update the belief over the current state. Additionally, we compare our approach with a particle filter algorithm proposed in Hong et al. (2020b) about non-stationary Latent Bandits. They consider two settings: one with complete knowledge of both the observation and transition models and another that incorporates priors on the parameters of the models to account for uncertainty. We compare against a mixture of these two settings by providing their algorithm with full information about the observation model (as it is for our case) and an informative prior about the true transition model. The comparison is made in terms of the empirical cumulative regret $\widehat{\mathfrak{R}}(t)$, which is the empirical counterpart of the expected cumulative regret $\mathfrak{R}(t)$ averaged over multiple independent runs.

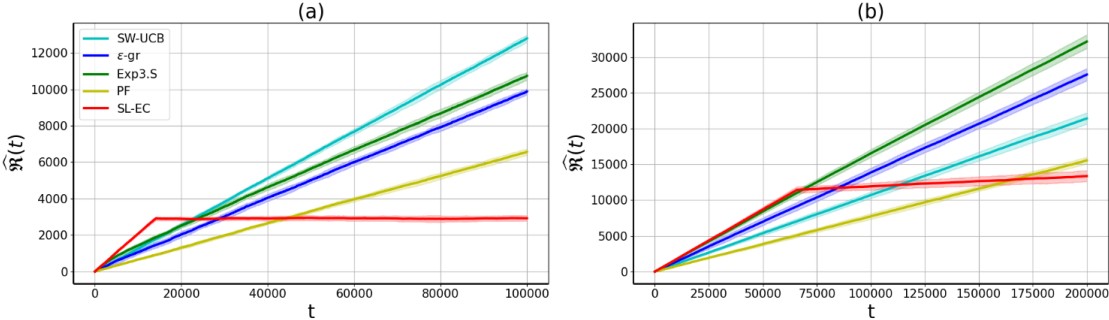

Figure 2: Plots of regret comparing the SL-EC Algorithm with some non-stationary bandit algorithms using: (a) synthetic data with parameters $S = 3$ states, $I = 4$ actions and $V = 5$ observations (5 runs, 95% c.i.); (b) data from MovieLens assuming $S = 5$ states, $I = 18$ actions and $V = 5$ observations. (5 runs, 95% c.i.).

### 6.2.1 Synthetic Experiments

These experiments have been conducted on various problem configurations with different numbers of states $S$, actions $I$, and observations $V$. The regret results for one configuration are shown in Figure 2(a). From the figure, it is clear that most of the baseline algorithms display a linear time dependence for the regret. This is expected since these algorithms do not take into account the underlying Markov Chain that governs the process. The particle filter algorithm, despite being given a good initial prior on the transition model, is unable to achieve the performance of SL-EC in the long run. Conversely, we can notice a quite different behavior for our algorithm that, in line with an Explore-Then-Commit approach, initially accumulates a large regret and then experiences a drastic slope change when the exploitation phase begins. The regret shown in each plot is the average over all the runs. For further information regarding the generation of the transition model and observation tensor, as well as the hyperparameters used for the baseline algorithms, please refer to Appendix A.

As a remark, our algorithm outperforms the others when the spectral gap $\beta$ of the chain is not close to zero. Indeed, if this is not the case, simple exploration heuristics such as $\epsilon$-greedy would lead to comparable performance. A clear example is when the transition matrix $\boldsymbol{P}$ defining the chain assigns equal probability to all transitions. In this scenario, all states can be considered independent and identically distributed, and we get no advantage from the knowledge of the matrix $\boldsymbol{P}$ over the use of an algorithm such as $\epsilon$-greedy.

### 6.2.2 MovieLens Experiments

We also perform some experiments on semi-synthetic data based on MovieLens 1M (Harper & Konstan, 2015), a well-known collaborative filtering dataset where users rate different movies each belonging to a specific set of genres. We adopt a procedure similar to the one used in Hong et al. (2020b). The dataset is initially filtered to include only users who rated at least 100 movies and the movies that have been rated by at least 100 users. After that, we combine the available information in order to obtain a table where each row contains the mean of the ratings for each observed genre for each user (user-genre-rating table). If the user didn't observe any movie belonging to a specific genre, the cell is empty. From the obtained matrix, we select 70% of all ratings as a training dataset and use the remaining 30% as a test set. The sparse matrices so obtained are completed using least-squares matrix completion (Mnih & Salakhutdinov, 2007) using rank 10 and leading to a low prediction error.

Having defined the appropriate rank, we use the predictions on the empty cells of the original *user-genre rating* matrix to fill the entire table. We define a switching bandit instance by using the notion of a *superuser* inspired by Hong et al. (2020b). We use $k$-means to cluster users using the rows of the user-genre-rating matrix. The users belonging to the same cluster define a superuser that embeds a set of users with similar tastes. The information about the users belonging to the same clusters is then combined and used to generate categorical distributions on the rating, given each superuser and each possible genre (our actions). We choose $k = 5$ for the number of superusers as it is the one that yields clusters with more similar dimensions and we

Table 1: Comparison with Nearly Deterministic Models

| 2 States | 3K samples | 6K samples | 9K samples | 15K samples |
|---|---|---|---|---|
| *SD O* | 0.0493 (0.0097) | 0.0379 (0.0103) | 0.0335 (0.0097) | 0.0259 (0.0081) |
| *SD T* | 0.0342 (0.0185) | **0.0189** (0.0097) | 0.0149 (0.0032) | 0.0101 (0.007) |
| ***OUR*** | **0.0234** (0.015) | 0.02 (0.0203) | **0.0119** (0.009) | **0.008** (0.0032) |
| 3 States | 150K samples | 300K samples | 600K samples | 900K samples |
| *SD O* | 0.0165 (0.0044) | 0.0113 (0.0036) | 0.0097 (0.0033) | 0.0085 (0.0018) |
| *SD T* | 0.1547 (0.0517) | 0.154 (0.0532) | 0.1544 (0.0534) | 0.1541 (0.0532) |
| ***OUR*** | **0.0066** (0.0026) | **0.0046** (0.0012) | **0.0037** (0.0018) | **0.0031** (0.0008) |
| 5 States | 150K samples | 300K samples | 600K samples | 900K samples |
| *SD O* | 0.0681 (0.0178) | 0.0513 (0.0111) | 0.0354 (0.0127) | 0.0283 (0.0082) |
| *SD T* | 0.2409 (0.0633) | 0.2484 (0.0584) | 0.243 (0.0603) | 0.2407 (0.0601) |
| ***OUR*** | **0.0283** (0.0054) | **0.0195** (0.0036) | **0.0137** (0.0033) | **0.0115** (0.0034) |

use $I = 18$ for the actions since it represents the number of identified genres. The number of observations $V = 5$ corresponds to the 5 possible ratings that a movie can get. The transition matrix that governs the dynamics with which superusers alternate is defined by giving higher probabilities to transitions to similar states and also giving higher weights to self-loops in order to avoid too frequent changes. The interaction goes as follows. At each step, a new superuser $s_t$ is sampled based on $s_{t-1}$ and the transition matrix. The agent chooses an action $a_t$ corresponding to a genre to propose and gets a rating that is sampled from the categorical distribution with parameters $\mathbf{O}_{s_t, a_t, :}$.

As for the synthetic case, our algorithm is compared to other baselines. From Figure 2(b), we can see that our SL-EC still outperforms the other baselines in the considered horizon. However, we highlight that our goal is not to beat the baselines since the comparison is not fair as most of them do not take into account the underlying Markov process, but we aim to show the difference w.r.t. other algorithms belonging to state of the art. More details about the experiments on Movielens can be found in Appendix A.

## 6.3 Comparisons with Modified Spectral Decomposition Techniques

The focus of this last set of experiments is to show the difference between a modified Spectral Decomposition (SD) technique and our approach. Among the various applications, SD techniques are typically used for learning with Hidden Markov Models (HMM) where no information about the observation and transition model is provided. In particular, Zhou et al. (2021) makes use of these techniques to get an estimation of both the observation and the transition model. It is important to highlight that SD methods are hardly used in practice because of their computational and sample complexity. Indeed, both the related works of Zhou et al. (2021) and Azizzadenesheli et al. (2016) include only proof-of-concept experiments with 2 hidden states and 2 possible actions.

To make the comparison fairer, since our algorithm requires knowledge about the observation model, we consider a modified SD technique in order to help the estimation process. The original SD technique to which we refer follows the procedures highlighted in Anandkumar et al. (2014) for HMM and makes use of the Robust Tensor Power (RTP) method for orthogonal tensor decomposition. In typical SD techniques, data is collected by sampling an action at each time step and updating the computed statistics with the observed realization. With the presented modified SD technique, at each step, we do not simply update the statistics with the realization observed given the pulled arm but we give information about the observation distribution for all the available arms, with this information being conditioned on the underlying current state. In this way, it is like pulling at each step all the arms and receiving full information about their associated reward distributions, given the underlying state.

We perform various experiments by fixing the number of arms ($I = 20$) and the number of possible rewards ($V = 5$) for each arm and by changing the number of states. Each experiment is performed over 10 different runs, where for each run a transition matrix and observation tensor is generated. For our algorithm, we selected for each experiment 3 arms among the 20 available using our offline arms selection strategy. The

Table 2: Comparison with Higher Model Stochasticity

| 2 States | 150K samples | 210K samples | 270K samples |
|---|---|---|---|
| *SD O* | 0.1500 (0.2639) | 0.1411 (0.2741) | 0.1455 (0.2665) |
| *SD T* | 0.1488 (0.1536) | 0.1699 (0.1742) | 0.1576 (0.1702) |
| ***OUR*** | **0.0145** (0.0175) | **0.0145** (0.0134) | **0.0125** (0.0103) |
| 3 States | 300K samples | 600K samples | 900K samples |
| *SD O* | 0.2987 (0.2128) | 0.3078 (0.2177) | 0.2594 (0.2309) |
| *SD T* | 0.3916 (0.2804) | 0.4425 (0.2637) | 0.4187 (0.2728) |
| ***OUR*** | **0.0077** (0.003) | **0.0063** (0.0023) | **0.0052** (0.002) |

transition and observation matrices are created in two different ways: we will see a first set of experiments (Table 1) where the two matrices are almost deterministic, hence having high probability on a specific observation/state and low probabilities for all the others. For transition matrices, the highest probability is assigned to the probability of staying in the same state. Near-determinism is defined with the objective of simplifying the problem by making states more distinguishable.

Table 1 is structured in the following way. It contains three different sets of experiments where each set is characterized by a different number of states. By fixing the number of states for the experiments, we show three rows on the table: the first one (indicated with *SD O*) contains the Frobenius norm error in the estimation of the observation matrix with the modified SD technique, the second row (indicated with *SD T*) represents the Frobenius norm error for the transition matrix with the modified SD technique, while the third row represents (indicated with *Our*) represents the Frobenius norm error for the transition matrix estimated with our algorithm. For each experiment, we report the mean Frobenius norm error over the 10 runs and one standard deviation between parenthesis. The information we report about the error in the estimated observation matrix with the modified SD technique has the only objective of giving more interpretability to the error in the estimated transition matrix (*SD T*). Of course, our modified technique allows better estimation of the observation matrix with respect to the standard one. What we are really interested in is the estimation error of the transition matrix through the two different methods, indeed this information is separated from *SD O* by a dashed line. We show in bold the experiments with lower estimation errors. By inspecting the results, it is clear that *Our* approach outperforms the modified SD technique in almost all the scenarios. Comparable results are only achieved in the case of 2 states.

We also provide a second set of experiments, where the generated matrices have less peaked distributions and higher stochasticity, for both the transition and the observation models (Table 2). The discrepancy between our approach and the modified SD technique is more evident in this scenario. This aspect can be justified by the theoretical comparison in Section 5.2 where we have seen that SD techniques have a higher dependency on the $\sigma_{\min}$ of the different matrices with respect to our approach. Thus, when matrices (in particular the observation ones) are more stochastic, their $\sigma_{\min}$ decreases and this aspect results in a more difficult estimation procedure. Furthermore, besides the lower performances, the SD technique requires higher computational power, and experiments with a higher number of states were not able to reach convergence. In particular, experiments with more states and with higher model stochasticity were not able to reach convergence with a number of samples of the order $10^5$ and, by increasing this number, there were memory space problems with the used hardware (Intel i7-11th and 16G RAM).

Again, we would like to emphasize that SD techniques are explicitly meant to work in a different setting, intrinsically more complex, where no information about either the transition or the observation model is provided. However, with this set of experiments we wanted to show that if instead we have knowledge about the observation model, directly using this information in the SD techniques does not lead to performances comparable to our approach.

# 7 Discussion and Conclusions

This paper studies a Latent Bandit problem with latent states changing in time according to an underlying unknown Markov Process. Each state is represented by a different Bandit instance that is unobserved by

the agent. As common in the latent Bandit literature, we assumed to know the observation tensor relating each MAB to the reward distribution of its actions, and by using some mild assumptions, we presented a novel estimation technique using the information derived from consecutive pulls of pairs of arms. As far as we know, we are the first to present an estimation procedure of this type aiming at directly estimating the probabilities of the state transitions encoded in the matrix $W$. We have shown that our approach is flexible as it allows choosing combinations of pairs of arms with non-uniform probability and is easy as it does not require specific hyperparameters to be set. We also provided some offline techniques for the selection of the best subsets of arms to speed up the estimation process. We analyzed the dependence of the parameters on the complexity of the problem and we showed how our estimation approach can be extended to handle models with continuous observation distributions. We used the presented technique in our SL-EC algorithm that uses an Explore-Then-Commit approach and for which we proved a $\mathcal{O}(T^{2/3})$ regret bound. We conducted a theoretical comparison with the work of Zhou et al. (2021) taking into account the difference between the two settings. The experimental evaluation confirmed our theoretical findings showing advantages over some baseline algorithms designed for non-stationary MABs and showing good estimation performances even in scenarios with larger problems.

We identified different future research directions for the presented work such as designing new algorithms that are able to exploit the flexibility in the exploration policy determined by the defined procedure, allegedly in an optimistic way. It may also be interesting to deepen the understanding of this problem when dealing with continuous reward models, trying to design optimal ways to discretize them in order to reach faster estimation performances. We could also consider the extension to the continuous state space setting (e.g., linearMDPs). Among the main challenges in this scenario, we consider the adoption of a different representation for the reference matrix that would otherwise not be computable with infinite states and the redefinition of the stationary transition distribution matrix. In such a case, it might be beneficial to directly estimate the feature functions by means of which the linear MDP is defined. Finally, it might be worth considering a contextual version of the proposed setting. According to the assumptions made, for example, whether the context is discrete or continuous or whether it is related or not to the latent state, this aspect may bring another dimension to the observation space. Redefining the reference matrix by also taking this feature into account will likely lead to more informative components and help the estimation procedure.

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
