# OpenReview forum: "Switching Latent Bandits"
_TMLR — Rejected by TMLR_

### Review · Reviewer_PTVy · 2023-07-23

**Summary Of Contributions:**

This paper proposes a new approach to the switching latent bandits problem, where an agent must solve multiple MAB problems (parameterized by a latent state) that evolve according to an underlying, unknown Markov chain.

The algorithm is an explore-then-commit algorithm that uses the exploration phase to estimate the transition matrix of the Markov chain. Then, in the exploitation phase, the algorithm maintains a belief over the current latent state then chooses the optimal action under the believed latent state.

The authors analyze the regret of their algorithm that is to my knowledge, asymptotically competitive with existing approaches. Finally, they empirically show that is outperforms previous methods such as approximate inference (particle filtering) on multiple numerical simulations.

**Audience:**

Yes

**Broader Impact Concerns:**

The paper is primarily theoretical, so I do not see any ethical implications of the work.

**Claims And Evidence:**

Yes

**Requested Changes:**

Overall, I see the paper as adequate as a theoretical contribution, as it is the first analyze-able algorithm for the studied setting without unrealistically strong assumptions. I do believe the practicality of the paper can be greatly improved with the following changes (though none are critical for acceptance):

(1) The authors should add a motivation for studying the switching latent bandits setting, by introducing some relevant application.

(2) The authors can do some experiments using real data. The experiments can still be semi-synthetic, by maybe taking a dataset such as MovieLens, and assuming a "super-user" that behaves according to multiple different users under some synthetic markov chain.

I also believe the authors could discuss the following extensions (which would also only strengthen the work and are not critical for acceptance):

(1) Can a similar algorithm be proposed if the latent state evolves as a linear MDP instead? Allowing the latent state to be continuous would greatly alleviate some practical concerns with the paper.

(2) Can the algorithm be extended to include context (whose distribution changes with the latent state)? I feel that this would only help with estimating the transition matrix, but it would be interesting to see the degree to which it may affect the current analysis.

**Strengths And Weaknesses:**

Strengths
---
The paper is organized clearly and well-written. The authors do a good job of motivating matrix estimation using stationary distributions before presenting their algorithm. The algorithm is also novel and requires fewer assumptions (on the transition matrix) than previous approaches. The analysis is also novel (particularly in deriving the estimation error of the transition matrix).

Weaknesses
---
It is unclear to me how limited the assumed setting of switching latent bandits is, particularly in assuming a discrete latent state. I agree that many realistic bandit problems are actually non-stationary (i.e. user preferences may change), but do not believe using a finite-state markov chain is the most realistic model of this non-stationarity. Furthermore, the experiments in the paper use purely synthetic data, so it is unclear how practical the proposed algorithm is in realistic scenarios (to my knowledge, explore-then-commit style algorithms are not that prevalent in practice).

---

> ### Author Response · Authors · 2023-08-30
>
> We thank the reviewer for their time and comments. We address the main questions and doubts highlighted in the weakness section.
> Regarding the use of discrete latent states, this choice can be considered valuable when approaching the modeling of complex real-life problems that are characterized by different and recurrent working regimes (for example those related to seasonality). These regimes can be typically observed in domains such as the financial market and online advertising where identifying and predicting future patterns can be of valuable interest. Regarding the use of a Markov Chain to model such processes, we highlight that it can be useful to model regimes with a specific duration. Indeed, the probability of remaining in the same state is modeled through a geometric distribution with the parameter being $(1 - p$) with $p$ being the probability of the self-loop for each state. Clearly, much more complex models (e.g., non-Markovian) could be considered, introducing likely new technical challenges.
>
> Concerning the requested changes, as suggested by the reviewer, we added in the Introduction section some examples of potential applications that can benefit from the use of our method. They are mainly related to online advertising and to the financial market. Regarding the experiments on real-world data, we followed the suggestion of the reviewer using the Movielens dataset. The results are reported in the experimental section.
> We also added in the last section some discussions about the proposed extensions. We agree with the reviewer that these extensions are of great interest, although we think they are out of the main scope of the present paper.

---

### Review · Reviewer_djC9 · 2023-08-13

**Summary Of Contributions:**

This paper considers a **latent bandit** problem, where the latent state follows an underlying Markov chain and each state is represented by a specific bandit instance. This paper also assumes that the learning agent is aware of the observation models of the arms conditioned on each latent state, but needs to learn the transition model of the underlying Markov chain. This paper formulates the considered problem in Section 3 and develops an algorithm, referred to as SL-EC Algorithm (Algorithm 1), in Section 4. A regret bound is established in Section 5 (Theorem 5.1), and preliminary experiment results are established in Section 6.

**Audience:**

Yes

**Broader Impact Concerns:**

None.

**Claims And Evidence:**

Yes

**Requested Changes:**

1) Please further motivate and justify the considered setting. Why not consider the more realistic and practical setting considered by Zhou et al. (2021)?

2) Please either prove the current $\tilde{O} \left( T^{2/3} \right)$ regret bound is tight (up to logarithmic factors) by deriving a matching lower bound, or further improve the current algorithm and/or regret bound.

3) Please try to strengthen the analysis by removing Assumptions 3.1 and 3.2. If the authors believe that these assumptions are essential, please further justify it.

**Strengths And Weaknesses:**

**Strengths:**

- The considered problem is interesting.

**Weaknesses:**

- The considered setting is restricted. In particular, I think a more realistic and practical setting is a setting where both the "conditional reward model" and the "state transition model" are unknown and need to be learned jointly, as considered by Zhou et al. (2021). Why not consider this more realistic and practical setting?

- Assumptions 3.1 and 3.2 are unnatural and probably unnecessary. My understanding is that they are made for technical reasons. For instance, my understanding is that Assumption 3.2 is made to recover the matrix **P**. However, do we really need to fully recover matrix **P** to derive a good regret bound? For instance, in degenerate cases where two states have the same bandit model (conditional reward model), this seems to be unnecessary.

- The regret bound in Theorem 5.1 might be further improved. In particular,

  - The dependence on $1/\pi_{\min}$ and $1/\sigma_{\min}(A)$ is undesirable, since they can be very large. Can we remove these terms in the regret bound?

  - The $\tilde{O} \left( T^{2/3} \right)$ regret bound might be undesirable. My understanding is that most of the state-of-the-art algorithms for bandit problems have $\tilde{O} \left( T^{1/2} \right)$ regret bound, and it is not clear to me if the $\tilde{O} \left( T^{2/3} \right)$ regret bound is tight. If the authors believe that this regret bound is tight, might they derive a matching lower bound? Otherwise, might the authors further improve this regret bound?

- The proposed **explore-then-commit** algorithm seems to be naive. I am not sure if a better regret bound (e.g. a $\tilde{O} \left( T^{1/2} \right)$ regret bound) can be derived under a different algorithm (e.g. an upper-confidence bound type algorithm).

- In the numerical simulations, for algorithms comparisons, the baseline algorithms do not have the knowledge of the observation tensor or the underlying Markov chain. Thus, I am not sure if the comparison is fair.

---

> ### Author Response · Authors · 2023-08-30
>
> We thank the reviewer for their time and comments. We try to address here the main issues pointed out in the weakness section.
> 1. **Restriction of the setting** The presented setting is not as restricted as it may appear. Indeed, in the real-world scenario, it is not unlikely to have large amounts of historical data that allow learning different working regimes beforehand. In this work, we are able to show that when this type of information is available, it is possible to get rid of classical spectral decomposition (SD) approaches and use a more favorable and efficient way to learn the transition matrix. It is also important to notice that using SD methods requires a huge amount of data even for small problems. We indeed highlight that both [1] and [2] that make use of SD techniques only provide proof-of-concept experiments with 2 hidden states and 2 possible actions, while our method is able to solve bigger problems, thus closer to more practical scenarios. For a more insightful discussion about this aspect, we point the reviewer to Section A.3 from the Appendix where we show a comparison between our method and a modified Spectral Decomposition technique that is provided with full information of the observation matrix. The simulations clearly show the advantage of using our approach in this scenario, thus justifying the method we propose.
> 1. **Assumptions** We start discussing Assumption 3.2. As already stated by the reviewer, this type of assumption is needed in order to learn the transition matrix. We believe that this assumption is fundamental in our setting. Indeed, similar assumptions are employed in many works dealing with learning in partially observable settings. It is utilized in [1] and [2] and, in the more general POMDP setting, it is also used to characterize a rich family of tractable POMDPs called $\alpha$-revealing POMDPs where the $\alpha$ value is a lower bound to the minimum singular value of the observation matrix, thus having $\sigma_{\min}(O) \ge \alpha > 0$ as done in [3, 4, 5].
>     We agree with the reviewer that in some subclasses of degenerate cases he pointed out and in similar ones, it may be not necessary to recover the transition matrix. However, we seek to address a more general class of problems which includes non-degenerate cases for which estimating the transition dynamics might be necessary to achieve low regret. Furthermore, we report that all the previously cited works employ techniques that aim at estimating the underlying models.
>     Concerning Assumption 3.1, it is mainly used for technical reasons. In particular, this term allows bounding the 1-norm error between the estimated belief and the real one, as can be observed in Proposition C.2 from the Appendix. We derived this assumption from [1] and in turn, they derive it from the work of [6] which is the first to derive this type of result. From their analysis, this assumption is needed in order to contrast the problems due to the normalization operation of the belief over the states that is done at each step.
> 1. **Dependence on the Parameters and Regret Bound** The dependence on $\pi_{\min}$ derives from bounding the error in the estimated transition matrix $P$ with the error of the stationary transition matrix $W$. The derived analysis is new and this dependence turns out to be necessary in our case. Regarding instead the dependence on $\sigma_{\min}(A)$, it is clearly strictly related to Assumption 3.2. This term always appears with similar dependence in related works[1, 2, 3, 4, 5]. As an example, concerning the more complex episodic POMDP scenario, [3] also provides a lower bound containing a $1/\alpha^2$ dependence for the regret with $\sigma_{\min}(O) \ge \alpha$. It is indeed quite likely that this dependence cannot be removed from the regret bound.
>     Talking about the regret bound, this result is typical in simple explore-then-commit approaches. For the moment there are no tight lower bounds for the considered continuous setting and we are currently unable to provide one.  It is still an open question whether this result could be improved. While we agree with the reviewer that a matching lower bound would close the problem, we take the liberty to point out that, in our view, closing this gap for a newly considered problem might be a too-demanding requirement for a single publication.
>
> Due to space limitations, we continue the discussion in the following comment.

---

> > ### Author Response · Authors · 2023-08-30
> >
> > 4. **Algorithms Comparison** We are aware that most of the baseline algorithms do not have information about the observation matrix and we make it clear in the experimental section. The knowledge of the observation matrix undoubtedly brings more information to the agent, but it is of minor help if the transition matrix is unknown. Our focus, in this case, is not on beating the baselines as the comparison is not fair but just to show the difference w.r.t. other algorithms belonging to state of the art. Among the considered baselines, the one using Particle Filter is the only one that assumes an underlying dynamic based on a Markov Chain. Differently from the other baselines, this algorithm is provided with full information for the observation matrix and an informative prior for the transition matrix.
> >
> > [1] Xiang Zhou, Yi Xiong, Ningyuan Chen, and Xuefeng Gao. Regime switching bandits. NeurIPS 2021.
> > [2] K. Azizzadenesheli, A. Lazaric, and A. Anandkumar. Reinforcement learning of POMDPs using spectral methods. arXiv preprint arXiv:1602.07764, 2016.
> > [3] Qinghua Liu, Alan Chung, Csaba Szepesvari, and Chi Jin. When is partially observable reinforcement learning not scary? In Proceedings of Thirty Fifth Conference on Learning Theory, volume 178 of Proceedings of Machine Learning Research, pages 5175–5220. PMLR, 02–05 Jul 2022.
> > [4] Chi Jin, Sham M Kakade, Akshay Krishnamurthy, and Qinghua Liu. Sample-efficient reinforcement learning of undercomplete POMDPs. NeurIPS, 2020.
> > [5] Liu, Q., Netrapalli, P., Szepesvari, C., and Jin, C. Optimistic mle–a generic model-based algorithm for partially observable sequential decision-making. arXiv preprint arXiv:2209.14997, 2022.
> > [6] Yohann De Castro, Élisabeth Gassiat, and Sylvain Le Corff. Consistent estimation of the filtering and marginal smoothing distributions in nonparametric hidden markov models. IEEE Transactions on Information Theory,2017.

---

### Review · Reviewer_tmxF · 2023-08-23

**Summary Of Contributions:**

The paper considers latent multi-armed bandit problems, where the latent state is not revealed (contextual bandits but the context is not given) and may change according to action-irrelevant probabilities. Under a fast-mixing of underlying states and the full-rank reward-emission probability assumptions, authors propose an explore-then-commit algorithm that achieves $T^{2/3}$ regret.

**Audience:**

Yes

**Broader Impact Concerns:**

This is a theoretical paper and there are no specific broader impact concerns.

**Claims And Evidence:**

Yes

**Requested Changes:**

- In Section 4.3, the authors discuss how to select a subset of arms when the number of arms is large. It would be good to be more precise: what is how detrimental, how do we implement the procedure to select $G$, etc

- I found a highly relevant recent work on latent bandits [1]. Authors are encouraged to discuss the differences in assumptions and techniques.

[1] Kwon, J., Efroni, Y., Caramanis, C., & Mannor, S. (2022). "Tractable Optimality in Episodic Latent MABs". Advances in Neural Information Processing Systems.

**Strengths And Weaknesses:**

Strengths

- The proposed problem is important, challenging, and under-explored.


Weakness

- Why it is necessary to recover all latent parameters perfectly? Wouldn't it suffice to match marginal trajectory distributions? The parameter matching guarantee necessarily suffers from the minimum mixing weights and singular values as in this paper.

- Why $\sqrt{T}$ regret is not possible? Do you have at least any intuition?

- I am not sure how this problem with the proposed assumptions is not tackled by the old paper on general POMDPs [1]. Authors should discuss clearly what is their additional challenge.

[1] K. Azizzadenesheli, A. Lazaric, and A. Anandkumar (2016), "Reinforcement learning of POMDPs using spectral
methods," in Conference on Learning Theory.

- Why in Assumption 1, all probabilities $> \epsilon$? Why isn't it enough to have $P$ to be irreducible and aperiodic?

---

> ### Author Response · Authors · 2023-08-30
>
> We thank the reviewer for their comments on our work. We address here the different points raised in the weakness section.
> 1. **Recovering All Latent Parameters** Many similar works addressing partially observable problems are model-based and aim at recovering the latent parameters describing the process. In a similar way, we also develop a model-based approach where recovering the full transition matrix results to be necessary in order to bound the regret. Indeed, in the general case, it is always possible to find situations where even a small error on the estimated transition matrix leads to different belief updates and potentially different pulled arms. As highlighted in the response to reviewer *djC9*, there are subclasses of this problem that may not necessitate the full estimation of the transition matrix, but this is not the case for general problems in this setting.
>     Regarding the question about matching marginal trajectory distributions, it is not clear how this can be done on any length trajectories without estimating the transition matrix. Furthermore, if the trajectories include the actions pulled at any steps, the marginals over trajectories should also depend on the policy used and this would make the estimation less sample-efficient. By instead directly estimating the transition matrix, we are able to decouple the policy from the underlying Markov process.
>     Concerning the dependence on specific parameters, we believe that some of them are unavoidable and characterize the difficulty of the problem. As an example, in similar partially observable settings, considering a specific subclass of POMDPs with episodic interactions, it has been shown ([3]) that the minimum singular value of the observation matrix appears in the expression defining the lower bound of the problem. We are currently not able to show this dependence in our specific setting but we strongly believe that a similar dependence also holds for our setting.
> 1. **Comparison with *Reinforcement learning of POMDPs using spectral methods*** There are several differences between our work and the one of [2]. The cited work assumes a more general POMDP setting where the pulled actions influence the dynamics of the process (as in classical MDPs) while in our case the latent process is a Markov Chain and the choice over the pulled actions will not change the sequence of underlying states. Furthermore, we assume to have knowledge about the observation tensor while they need to estimate it.
>     This last difference in the assumptions leads to the first difference in the used approach. In order to estimate the transition and observation matrices, they use classical spectral estimation techniques while, in our case, we exploit the knowledge of the observation tensor to learn the transition matrix using our newly designed estimation technique. Another major difference characterizing the two works relies on the used oracle. Differently from our approach that builds a belief over the subsequent state using all the historical information, their oracle is the optimal memoryless policy, i.e. a markovian policy that only depends on the current observation instead of using all past observations to form the belief of the underlying state. Using this type of policy allows avoiding the introduction of the belief state. In general cases, the performance gap between their oracle and ours is linear in T. This is definitely the additional challenge we face with respect to their work.
> 1. **Order of the Regret** We are currently not able to provide a lower bound for this setting and we are not aware whether this result on the regret can be improved. Concerning similar related works, some of them have been able to provide $\sqrt{T}$ regret but either under weaker oracles (optimal memoryless policy [2]) or by considering different settings (such as the episodic one). Strictly related to this last case, we consider the work on revealing POMDPs [3] where the interaction is episodic and they use as oracle the optimal policy of the POMDP. They are able to provide a regret guarantee that scales with $\sqrt{K}$ with $K$ being the number of episodes but with a polynomial dependence on the horizon $H^\beta$ of each episode, with $\beta \ge 3$. For the non-episodic setting instead, the related work of [1] achieves $T^{2/3}$ regret and, to the best of our knowledge, there are no other works with a setting similar to ours showing $\sqrt{T}$ regret.
>
> Due to space limitations, we continue the discussion in the following comment.

---

> > ### Author Response · Authors · 2023-08-30
> >
> > 4. **Discussion on Assumption 3.1** This assumption is imported from the similar work of [1] and is mainly used for technical reasons as appears in Proposition C.2 in the appendix. It is needed to bound the 1-norm error between the belief computed using the estimated transition matrix and the one computed using the real one. This assumption in turn derives from the work of [4] that is the first to derive this type of result. From their analysis, this assumption is needed in order to contrast the problems due to the normalization operation over the states which is done at each step.
> >
> > Regarding the requested Changes, we give more details about the selection of the subset of arms and discuss the differences with the related work cited by the reviewer.
> > 1. We thank the reviewer for the suggestion. We inserted more details in Section 4.3. We provide here an additional discussion on the subject.
> >     In the case of a large number of arms, using a round-robin procedure may be detrimental as it considers all arms equivalently. There may be cases where some actions carry less information. The extreme case is an action that induces the same observation distribution for all the switching MABs. Indeed, pulling that action will not provide any additional information on which is the current MAB and the effect will only be to slow down the estimation procedure. In general, actions that induce *similar* observation distributions for all the MABs will provide *less* information with respect to actions that induce highly different distributions for all the MABs. All this information is encoded into the $\sigma_{\min}(A)$. Concerning instead the construction of the $G$ matrices when the number of arms to use $J$ is given, we can say the following. The idea is to consider all the possible combinations of the subsets of dimension $J$ of the original arms and for each combination, select from the original reference matrix $A$ only the rows associated with couples of arms that appear in the subset. Intuitively, for each generated subset, this procedure corresponds to redefining new simplified MAB instances having as actions only those appearing in the subset. From these new reduced MABs, the standard procedure is used to construct the new reference matrix $G$. Having defined reference matrices for each generated subset, their minimum singular values are compared and the subset of arms yielding a reference matrix with higher $\sigma_{\min}(G)$ will be chosen. As a last point, we would like to clarify that this offline selection procedure is only related to the exploration phase and after committing, all the available arms can potentially be used.
> > 1. Regarding the distinction with the work of [5], we discussed some aspects in the section devoted to related works. We summarize here the setting and the main differences with respect to our work. They assume an episodic setting with a fixed horizon $H$ where at the beginning of each episode a specific MAB instance is sampled from a fixed mixing distribution. The agent is unaware of which MAB he is interacting with. The goal is to learn both the mixture weights and the reward distributions associated with each MAB. Furthermore, they use a Wasserstein metric to measure the parameter distance between two latent models. There are several differences with respect to our work.
> >     - Reward distributions of each MAB have to be learned while we assume to know them.
> >     - In [5], MABs are sampled independently at the beginning of each episode, while we assume a dependence between MABs based on a Markov Chain.
> >     - They consider an episodic setting with a fixed horizon with the MAB being fixed during the whole episode, while our setting is continuous and the transition step between MABs happens every time step.
> >     - Concerning the results, they provide guarantees in terms of sample complexity, while we employ a regret analysis for the proposed setting.
> >
> > [1] Xiang Zhou, Yi Xiong, Ningyuan Chen, and Xuefeng Gao. Regime switching bandits. NeurIPS 2021.
> > [2] K. Azizzadenesheli, A. Lazaric, and A. Anandkumar. Reinforcement learning of POMDPs using spectral methods. arXiv preprint arXiv:1602.07764, 2016.
> > [3] Qinghua Liu, Alan Chung, Csaba Szepesvari, and Chi Jin. When is partially observable reinforcement learning not scary? In Proceedings of Thirty Fifth Conference on Learning Theory, volume 178, pages 5175–5220. PMLR, 02–05 Jul 2022.
> > [4] Yohann De Castro, Élisabeth Gassiat, and Sylvain Le Corff. Consistent estimation of the filtering and marginal smoothing distributions in nonparametric hidden markov models. IEEE Transactions on Information Theory,2017.
> > [5] Jeongyeol Kwon, Yonathan Efroni, Constantine Caramanis, and Shie Mannor. Tractable optimality in episodic latent mabs. In S. Koyejo, S. Mohamed, A. Agarwal, D. Belgrave, K. Cho, and A. Oh (eds.), Advances in Neural Information Processing Systems, volume 35, pp. 23634–23645. Curran Associates, Inc., 2022.

---

### Decision · Action_Editor_k557 · 2024-02-09

**Recommendation:** Reject

**Comment:**

I am a new Action Editor on this paper. The former Action Editor asked the authors to revise the paper based on their comments and then was changed. I made the following agreement with the authors:

* I will look at the paper myself and make decision about acceptance.

Unfortunately, I cannot recommend acceptance at this time. The main reason is that the paper is full unclear statements and ambiguities. Examples are:

Section 3.1: $S$ denotes both a random state and the number of states.

Equation 2: $d$ needs to be related to $a$ and $r$. How about $d = a V + r$?

Equation 4: Define $\sigma_{\min}$ and $\sigma_{\min}^2$.

Equation 5: Properly quantify $s'$ and $s''$.

Equation 5: $Q$ should be used as $Q(r_t \mid s', a_t)$ and $Q(r_t \mid s'', a_t)$.

Section 4.1: Exploration policy $\theta$ is not clearly defined. Based on (7), it seems to be a joint probability distribution over two consecutive rounds.

Equation 7: $P$ does not follow the earlier notation.

Section 4.1: (8) is computed using $D$, which is never formally defined.

Section 4.1: $\hat{w}$ in (9) is computed using $\hat{D}$, which is never formally defined.

Algorithm 1: SL-EC uses $\hat{P}$. Computation of $\hat{P}$ from $\hat{w}$ is described in the last paragraph of Section 4.1 and is ambiguous.

Algorithm 1: Line 10 is ambiguous. Why not $n \gets n + \dots$ instead of "Update $n$ with"?

Algorithm 1: Line 11 is ambiguous. Why not $D \gets D \cup \dots$ instead of "D.add()"?

Section 4.3: $\hat{P}_k$ is never properly defined.

I wanted to stress that I am less worried about theory. The proposed algorithm is explore-then-commit, and its analysis essentially follows from random initial exploration and non-adaptive concentration results.

**Audience:**

This paper will have audience among bandit researchers. It pushes the boundary of dealing with latent variables that change over time, which is both challenging and interesting.

**Claims And Evidence:**

The writing of the paper is ambiguous and needs to be improved. Since this is a second round of comments from an Action Editor, I suggest a major revision.

**Resubmission Of Major Revision:**

The authors may consider submitting a major revision at a later time.